# Studying nucleic envelope and plasma membrane mechanics of eukaryotic cells using confocal reflectance interferometric microscopy

Vijay Raj Singh [1,2,3], Yi An Yang[4], Hanry Yu [2,4,5,6], Roger D. Kamm [2,3,7], Zahid Yaqoob[1] & Peter T.C. So[1,2,3,7]

Mechanical stress on eukaryotic nucleus has been implicated in a diverse range of diseases including muscular dystrophy and cancer metastasis. Today, there are very few non-perturbative methods to quantify nuclear mechanical properties. Interferometric microscopy, also known as quantitative phase microscopy (QPM), is a powerful tool for studying red blood cell biomechanics. The existing QPM tools, however, have not been utilized to study biomechanics of complex eukaryotic cells either due to lack of depth sectioning, limited phase measurement sensitivity, or both. Here, we present depth-resolved confocal reflectance interferometric microscopy as the next generation QPM to study nuclear and plasma membrane biomechanics. The proposed system features multiple confocal scanning foci, affording 1.5 micron depth-resolution and millisecond frame rate. Furthermore, a near common-path interferometer enables quantifying nanometer-scale membrane fluctuations with better than 200 picometers sensitivity. Our results present accurate quantification of nucleic envelope and plasma membrane fluctuations in embryonic stem cells.

[1] Laser Biomedical Research Center, G. R. Harrison Spectroscopy Laboratory, Massachusetts Institute of Technology, Cambridge, MA 02139, USA. [2] Singapore-MIT Alliance for Research and Technology, Singapore 138602, Singapore. [3] Department of Mechanical Engineering, Massachusetts Institute of Technology, Cambridge, MA 02139, USA. [4] Mechanobiology Institute (MBI), National University of Singapore, Singapore 117411, Singapore. [5] Department of Physiology, National University of Singapore, Singapore 117593, Singapore. [6] Institute of Bioengineering and Nanotechnology, Agency for Science, Technology and Research, Singapore 138669, Singapore. [7] Department of Biological Engineering, Massachusetts Institute of Technology, Cambridge, MA 02139, USA. Correspondence and requests for materials should be addressed to Z.Y. (email: zyaqoob@mit.edu) or to P.T.C.S. (email: ptso@mit.edu)

The mechanical environment plays important role in many cellular physiological and pathological processes[1]. The impact of drug response on cellular biomechanics has also been highlighted as an important consideration in mechan-opharmacology[2–4]. Atomic force microscopy (AFM)[5] and optical tweezers-based methods[6] can provide direct measurement of the cellular plasma membrane elasticity. However, these methods are contact based and are perturbative where the measurement itself may elicit cellular responses. Particle-tracking microrheology can allow measuring mechanical forces on the plasma membrane, as well as within the cytosol by passively observing thermally driven motions of embedded tracer particles[7]. However, the delivery of particles into cells is often invasive, and the measurement results may depend on how tracer particles are processed by the cells after delivery. In summary, most existing methods that are capable of quantifying whole cell biomechanics are often perturbative and low throughput with very limited capability to quantify mechanical properties of thousands to millions of cells on a population level.

It is also impotant to note that most existing methods provide overall mechanical characteristics of the cell[8,9] with limited ability to resolve the relative contributions from the plasma membrane, the cytosol, or the internal organells. Among cellular organelles, nucleus is the largest and stiffest component, exposed to extracellular and intracellular mechanical forces. Recently, the importance of nuclear mechanics has been recognized due its strong correlation with gene signaling and gene transcription[10,11]. Nuclear stiffness has also been shown to regulate cell phenotype during the stem cell differentiation[12]. Furthermore, the extra-cellular forces can also modify nuclear shape, structure, and stiffness[13,14] that may have important role in cellular physiological and pathological processes[15,16]. For instance, nuclear pore selectivity and loss of nuclear envelope integrity are shown to be linked with various human diseases including cancer[17–21]. Hence, biomechanics-related properties of the nucleus, e.g., stiffness, disorder, compactness, and deformability need to be quantified accurately for studies involving nuclear mechanotransduction. However, non-invasive and accurate measurement of nucleic mechanical properties is challenging. For example, partial wave spectroscopy provides change in characteristic length of spatial fluctuations of refractive index of the nucleus, called disorder strength[22]. The disorder strength is related to the macromolecular organization; however, it does not directly inform on the mechanical properties of the nucleus itself. Recently, quantitative phase microscopy (QPM)-based methodologies have also been presented to estimate the disorder strength, which can be related to the refractive index variance in biological cells and tissue samples. More precisely, the measured disorder strength parameter has been linked with the cell stiffness[23] and used to compare benign versus malignant breast tissue biopsies[24]. Interestingly, Brillouin spectroscopy enables non-invasive quantification of material properties by quantifying the interaction of light with spontaneous acoustic phonons, providing longitudinal modulus of the material in the gigahertz (GHz) frequency range[25]. Further, Brillouin light scattering has also been reported to measure intracellular longitudinal modulus[26]. While this is a powerful method to determine relative stiffness of materials, the interpretation of nuclear deformation events on physiologically much shorter frequency scales based on modulus measured on the GHz scale requires many assumptions that are difficult to validate for complex fluids, such as the constituents of the nucleus and cytosol. In addition, Brillouin spectroscopy also requires point scanning in 3D, which renders the approach low throughput.

An important alternative approach that may provide quantitative mapping of nucleic mechanical properties is based on measuring nanometer scale thermally driven cell membrane fluctuations with millisecond temporal resolution using QPM[27]. Coupled with appropriate continuum or finite element mechanical models, the measured membrane fluctuations can further allow quantification of cellular elasticity and loss moduli[27]. This approach has been successfully utilized in staging various red blood cell (RBC) disorders[28–31]. However, since most QPM techniques are designed with transmission geometry with limited to no depth resolution, successful applications of the approach in quantitative biomechanical measurements are mostly confined to the studies of blood disorders involving imaging of RBCs with homogenous intracellular environment. For eukaryotic cells, depth-sectioning of about 1–2 μm is often needed to separate signal from the plasma membrane versus the nucleic envelope and vice versa. Realizing that studying eukaryotic cells with complex intracellular structure is critical to expand the utility of QPM in cell biomechanics, several generations of reflection-mode QPMs have also been developed with varied depth-sectioning capability[32–40] (Table 1). Since the refractive index contrast between the cytosol and the nucleus can be less than one part in 10,000, high-sensitivity QPM characterized by phase-stabilized interferometer design and low shot-noise-limited detection is required. We note, however, while some of these systems offer desired depth sectioning, none of them have sufficient sensitivity to quantify the nanometer-scale membrane fluctuations of the NE.

Here, we present a confocal reflectance interferometric microscope system that features 1.5 microns depth resolution and better than 200 pm height measurement sensitivity for high-speed (~70 Hz wide-field) characterization of nanometer scale nucleic envelope and plasma membrane fluctuations in complex eukaryotic cells. A theoretical model is presented for recovery of the correct displacement of the fluctuating membrane in the vicinity of the strong reflector. Measurements of nucleic envelope and plasma membrane fluctuations are carried out for mouse embryonic stem (ES) cells; the quantitative analysis shows higher fluctuation amplitude for plasma membrane than that for the nucleic envelope. Further, we also observe the nucleic envelope fluctuations of cells depend on their substrate rigidity.

## Results

**Confocal reflectance interferometric microscopy.** The core instrument innovation (Fig. 1) includes the use of confocal

### Table 1 Summary of depth-resolved reflectance QPM systems

| Depth-sectioning strategy | Illumination scheme | Common-path geometry | Best axial resolution (μm) | Best speed |
|---|---|---|---|---|
| Broad band source[32–35] | Point-scan, line-scan, wide-field | Yes[32–34] | ~4 | 1 kHz |
| Swept-source[36] | Wide-field | Yes | ~5 | 68 s |
| White light based[37] | Wide-field | No | ~1 | <1 Hz |
| VCSEL array-based[38] | Wide-field | No | ~8 | 10 kHz |
| Confocal slit-aperture[39] | Line-scan | No | ~2 | 20 Hz |
| Dynamic speckle-based systems[40] | Wide-field | No | ~1 | 100 Hz |

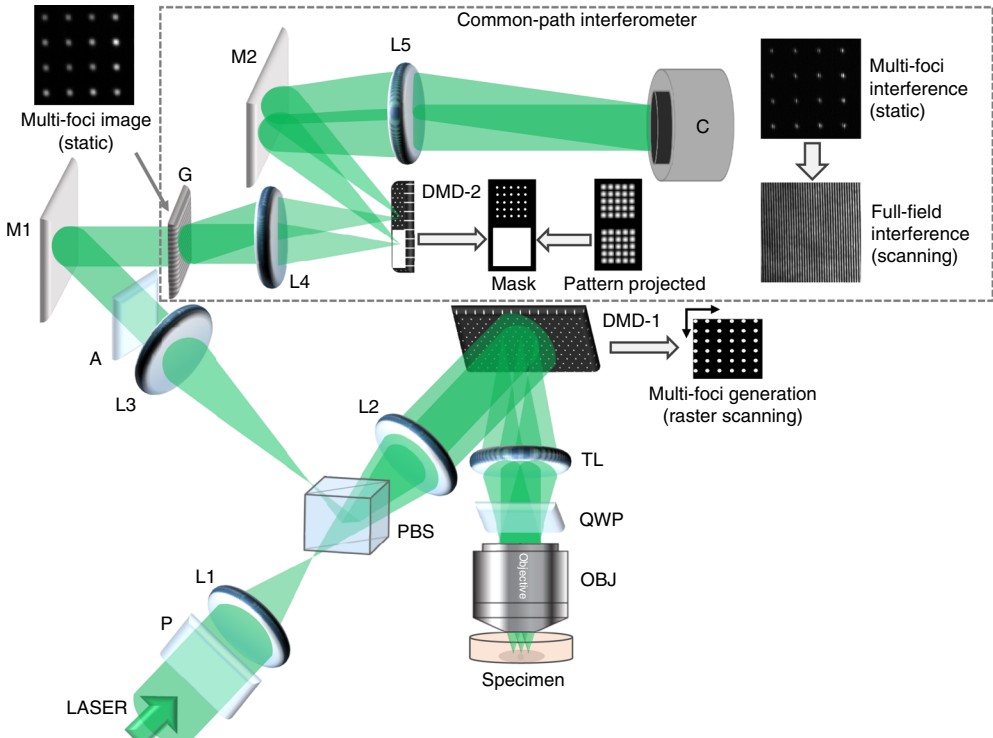

**Fig. 1** Schematic of the confocal reflectance interferometric microscope. Digital micro-mirror device, DMD-1, is used to create scanning rectangular array of pinholes for high-speed (millisecond frame rate) confocal imaging of the specimen. Detection of depth-resolved optical phase is achieved using the common-path interferometer. P polarizer, L1–L5 convex lenses (L1 = L2 = L3 = 300 mm, L4 = 100 mm, L5 = 500 mm), PBS polarizing beam splitter, DMD digital micro-mirror device, TL tube lens (165 mm), QWP quarter wave plate, OBJ microscope objective (Zeiss, × 40×/1.2), A analyzer, M1–M2 mirrors, G grating (80 lines pairs/mm), and C Camera

principle to achieve micron-level depth resolution and a high-speed digital micro-mirror device (DMD)[41] to generate a scanning confocal spots grid, allowing fast full-field imaging. A complete theory of the proposed confocal interferometric system is described in Supplementary Note 1. Finally, a highly stable near common-path interferometer, which shares the design with previously published diffraction phase microscopy[42], is implemented in the detection arm to quantify the confocal signal with height measurement accuracy better than 200 pm (Supplementary Note 2). A series of wide-field interferograms from an interface, such as the plasma membrane or the nucleic envelope, are acquired in an off-axis geometry. The measured single-shot off-axis interferograms can be analyzed to compute the phase maps $\phi(x,y;t)$ using Hilbert transform[43]. Furthermore, the phase maps corresponding to different cellular interfaces can be determined independent of each other if their separation is larger than the instrument's axial resolution.

For the proposed system, reference field is generated using the optical signal from the highly reflected glass interface, located at an off-focal plane, underneath the cell. This mechanism provides a phase-stabilized reference, which is critical for desired phase measurement stability. However, the presence of signal from glass interface, which is crucial to generate a stable reference, also suppresses the measured phase associated with the fluctuating membrane when computed using Hilbert transform. To this end, a method to characterize the effect of optical signal from off-focal plane glass interface on the measured phase and an approach to obtain correct phase reconstruction is presented in Supplementary Note 3. The corrected phase maps can be converted to height maps using the relation $h(x,y;t) = \frac{\phi(x,y;t) \times \lambda}{4\pi \times n}$, where $\lambda$ is the wavelength of the source and $n$ is the refractive index of the culture medium. The instantaneous cell membrane deformation map is obtained by subtracting its mean height from the instantaneous height reconstructed at each time point, i.e., $\Delta h(x,y;t) = h(x,y;t) - \langle h(x,y) \rangle$.

**Characterizing nucleic envelope and plasma membrane fluctuations.** To demonstrate the capability of the proposed optical system, we have characterized nucleic envelope and plasma membrane fluctuations in mouse ES cells, E14 (ATCC, CRL-1821). Figure 2a shows an interferogram recorded at the cell–dish interface (Z = 0 μm). Optical signal detected from the selected cell's bottom region is primarily dominated by the cell–dish interface reflections. While scanning the focal plane, back-scattered optical signal from nucleic envelope was observed at ~4 μm above the cell bottom surface. However, since the refractive index contrast is significantly lower for nucleic envelope interface, approximately seven times higher laser power was required for interferogram recording with similar SNR as that for the cell–dish bottom interface. Furthermore, the signal from the plasma membrane was observed at ~9 μm away from the bottom surface. The first row in Fig. 2b shows the zoomed region of interferogram in Fig. 2a, when microscope objective's focal plane coincides with the dish–cell membrane, the nucleic envelope, and the plasma membrane interfaces, respectively. Regions of nucleic envelope and plasma membrane are highlighted using red-dotted circles, and marked for further analysis. The corresponding 3D height maps reconstructed from the interferograms are shown in the bottom row of Fig. 2b. Animations of the out-of-plane fluctuations corresponding to the bottom interface, nucleic envelope, and plasma membrane are shown in Supplementary Note 4. To compare nucleic and plasma membrane fluctuations, plots of instantaneous fluctuation amplitudes, $\Delta h(x,y;t)$, at single spatial location of cell–dish interface, as well as for nucleic

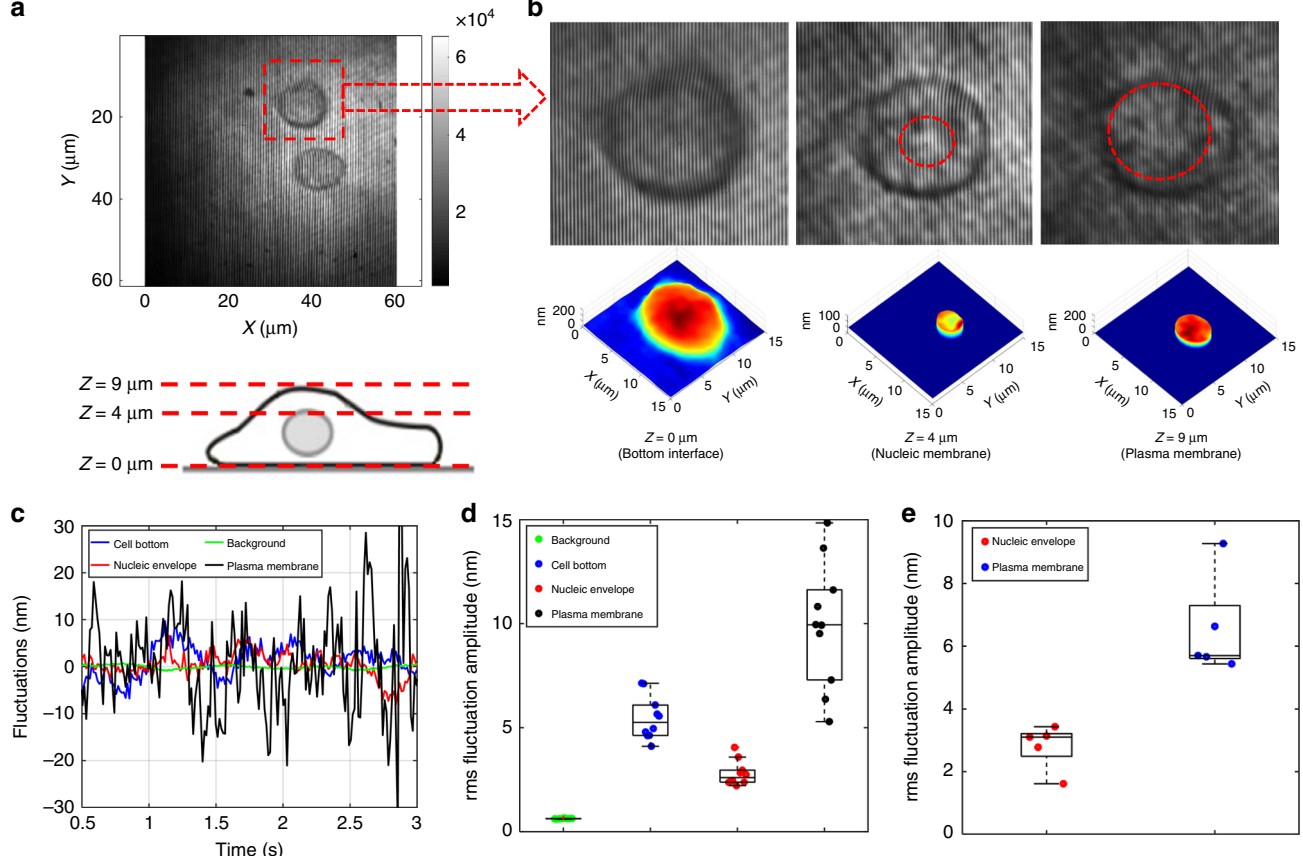

**Fig. 2** Nucleic envelope and plasma membrane fluctuations in embryonic stem cells. **a** Interferogram recorded at the cell–dish interface ($Z = 0\ \mu m$). Bottom cartoon shows three axial positions, cell–dish interface, nucleic envelope, and plasma membrane, where interferograms are recorded. **b** The upper row shows the zoomed interferogram at the corresponding axial positions, the red circles mark the nucleic envelope and plasma membrane. The bottom row shows the 3D height map reconstructed from the corresponding interferograms. Nucleic envelope and plasma membrane are masked in the 3D plot. **c** Amplitude of membrane fluctuations for background (outside cell region), cell bottom, nucleic envelope, and plasma membrane. **d** rms fluctuation amplitude for different spatial locations of corresponding interface of selected stem cell. The line within each box represents the median, and the lower and upper boundaries of the box indicate the first and third quartiles, respectively. Error bars (whiskers) represent the interquartile range ($n = 10$). **e** Plot of the rms fluctuation amplitude of nucleic envelope and plasma membrane for different embryonic stem cells. The line within each box represents the median, and the lower and upper boundaries of the box indicate the first and third quartiles, respectively. Error bars (whiskers) represent the interquartile range ($n = 5$)

envelope and plasma membrane interfaces are shown in Fig. 2c. Fluctuations of nucleic envelope and plasma membrane, represented by red and black traces, respectively, are clearly distinguishable. Similar measurements were also performed at multiple spatial locations of respective interfaces; the root-mean-square (rms) fluctuation amplitudes are plotted in Fig. 2d. Specifically, we found that the nucleic envelope median rms fluctuation amplitude (~2.59 nm) was smaller than that of the plasma membrane (~9.93 nm) indicating higher stiffness of nucleic envelope. We also observed suppressed fluctuations (represented by blue trace of Fig. 2c) at the cell bottom interface, which is likely due to multiple focal adhesions between the cell membrane and the substrate. Furthermore, the noise floor of the system (represented by green trace) was determined by analyzing a region outside the cell, i.e., at cell culture medium–dish interface and was found to be ~0.62 nm. Moreover, the analysis of nucleic envelope and plasma membrane fluctuations was also extended to multiple cells. The rms fluctuation amplitudes for nucleic envelope and plasma membrane for five different cells are shown in Fig. 2e, highlighting cell-to-cell variability. Note that each data point represents the mean rms fluctuation amplitude corresponding to multiple spatial locations on nucleic envelope or plasma membrane interface of respective cell.

The relaxation dynamics of nucleic envelope and plasma membrane fluctuations can be modeled by exponential decay phenomenon and characterized by the non-normalized temporal autocorrelation function (covariance) of the instantaneous height fluctuations. Specifically, we calculate the temporal autocorrelation function defined as

$$G(\tau) = \langle (h(t) - h)((h(t + \tau) - h) \rangle.$$

for different interfaces, where $\tau$ is the lag time. Figure 3a shows the autocorrelation function plot at a single $(x,y)$ location of corresponding interfaces. Variance of fluctuation amplitudes (zero lag time) was also calculated for multiple spatial locations of corresponding membranes; Fig. 3b shows measured variance, subject to the white noise, at multiple locations. Moreover, autocorrelation function for $\tau > 0$ represents the dynamics of membrane height fluctuations. For accurate variance of fluctuation amplitudes, autocorrelation function value is extrapolated to $\tau > 0$ time point by using spline fitting the rest of the autocorrelation plot. The corresponding corrected autocorrelation plots for cell bottom, nucleic envelope, and plasma membrane interfaces are shown in Fig. 3c. The variance of fluctuation data from the extrapolated autocorrelation function is shown in Fig. 3d. Clearly, the extrapolated autocorrelation

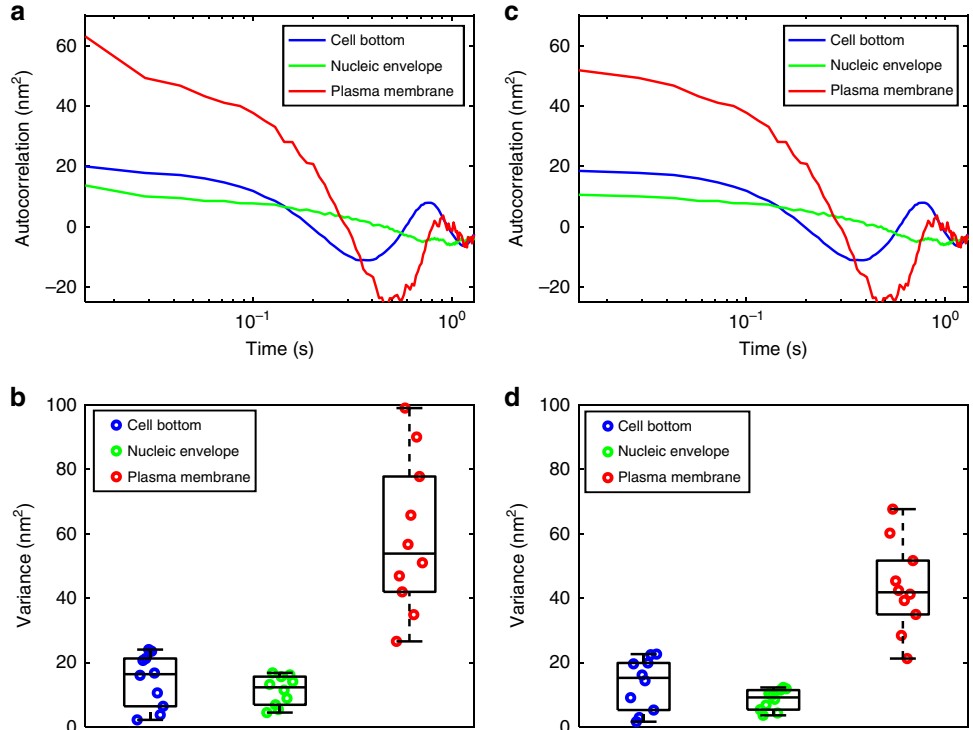

**Fig. 3** Temporal autocorrelation and variance analysis of nucleic envelope and plasma membrane fluctuations. **a** Original autocorrelation plot for cell bottom, nucleic envelope, and plasma membrane at a single spatial position of the corresponding interface. **b** Dot plot of variance of membrane fluctuations measured at multiple spatial locations based on the autocorrelation plot shown in **a**, which is the correlation value obtained at the 0-time point. The line within each box represents the median, and the lower and upper boundaries of the box indicate the first and third quartiles, respectively. Error bars (whiskers) represent the interquartile range ($n = 10$). **c** Extrapolated autocorrelation plots, where white noise at the zero-time point has been corrected. **d** Variance of membrane fluctuations measured at multiple spatial locations based on the extrapolated autocorrelation plots **c**. The line within each box represents the median, and the lower and upper boundaries of the box indicate the first and third quartiles, respectively. Error bars (whiskers) represent the interquartile range ($n = 10$)

suppresses the white noise from the fluctuations data and improves measurement accuracy for height fluctuation statistics.

To further demonstrate the ability of the proposed confocal reflectance interferometric microscope, we have quantified the effects of substrate rigidity on nucleic envelope fluctuations. In the past, studies based on fluorescent probes have indicated that nuclear mechanics is regulated by biophysical signals, such as externally applied forces through the manipulation of substrate rigidity[14]. Attempts have also been made to measure the temporal changes in the fluorescently labeled nucleic envelope[44]. We use our label-free approach to quantify the effect of substrate rigidity on the nucleic envelope fluctuations in a non-contact manner. Mouse ES cells E14 ES (ATCC, CRL-1821) were cultured, and undifferentiated cells were seeded at a moderate density (~$10^3$ cells/cm$^2$) on two Petri dishes, coated with fibronectin and gelatin substrates, respectively. After the cells had been seeded for 2 hours, nucleic envelope fluctuations were measured for both populations. Figure 4a shows the comparison of nucleic envelope instantaneous fluctuation amplitude at a single spatial location for two representative cells, one from each population. Clearly, reduced nucleic envelope fluctuations were measured for the cells adapted on fibronectin substrate versus gelatin. This may be due to the fact that cells make fewer focal adhesions on gelatin versus fibronectin[45]. A statistical analysis was also performed by analyzing multiple spatial locations of the nucleic envelope on five different cells, each coded in different color. Figure 4b shows out-of-plane nanometer scale rms fluctuation amplitude of the nucleic envelope with milliseconds temporal resolution for different cells adapted on fibronectin and gelatin substrate. In comparison, the

previous fluorescence-based study[44] reports nucleic envelope fluctuations with diffraction-limited spatial resolution, and temporal resolution on the order of seconds.

## Discussion
In the past, efforts have been made to estimate RBCs' membrane rheological properties using nanometer scale membrane fluctuations information measured using diffraction phase microscopes[29,46,47]. These studies have led to quantification of RBC pathophysiology in malaria[28], metabolic remodeling of human RBC membrane[48], and RBC stiffness in sickle cell disease[49]. Specifically, the theoretical framework[47,50] interprets the measured membrane fluctuations using a viscoelastic continuum model of the composite spectrin-network/lipid membrane bounded by bulk viscous fluids on both sides. While this is a good model for RBCs, it is clearly insufficient for eukaryotic cells. Unlike RBCs, the interior of eukaryotic cells is not homogenous but has at least two major compartments, namely, the cytosol and the nucleus, with different viscoelastic properties[51,52]. Therefore, the simplest model must consists of three compartments: the medium (pure viscous fluid), the cytosol (viscoelastic medium), and the nucleus (a different viscoelastic medium) bounded by two interfaces (the plasma membrane and the nucleic envelope) with different bending, $\kappa$, and extension, $\sigma$, moduli. Granek[53] has reported a relevant model that can recover bending modulus of a membrane driven by thermal forces separating two viscoelastic media with different frequency-dependent complex moduli using spatial and temporal autocorrelation functions of the interface

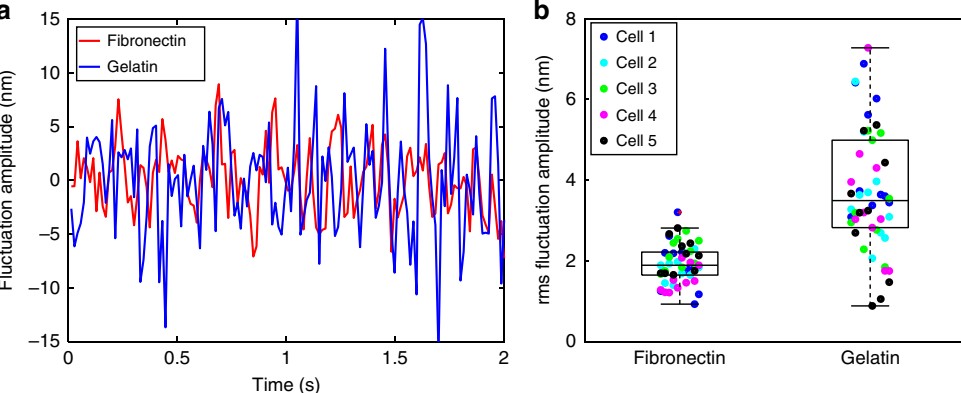

**Fig. 4** Role of extracellular matrix stiffness in nucleic envelope fluctuations. **a** Comparison of nucleic envelope fluctuation amplitude at a single spatial location for two embryonic stem cells, adapted on fibronectin and gelatin substrates, respectively. **b** Dot plot of rms fluctuation amplitude for the two stem cell populations. This analysis was achieved by considering multiple spatial locations of the nucleic envelope on five different cells, coded in different colors, for each population. The line within each box represents the median, and the lower and upper boundaries of the box indicate the first and third quartiles, respectively. Error bars (whiskers) represent the interquartile range ($n = 50$)

fluctuations that can be quantified using confocal phase microscopy presented here. Since we expect that extension moduli to dominate in the spatial/temporal scales of eukaryote cells measured in millisecond to second time scale, the Granek model will need to be extended theoretically to account for both the bending and stretch moduli; this extension is fairly simple given these two moduli have very different spatial and temporal power dependence. Subsequently, the Granek model can then be applied for this three-compartment case to determine plasma and nucleic membrane moduli assuming the viscoelastic properties of the cytosol and the nucleoskeleton can be independently measured. Recent development of force spectrum microscopy[54], an advanced form of laser trap-activated particle rheology, offers an attractive approach to accurately determine the unknown frequency-dependent viscoelastic properties of these two compartments. Our future experiments are planned to incorporate force spectrum microscopic measurement in model systems to validate the recovery of plasma and nucleic membrane bending and extension moduli based on confocal phase microscope measurements of their fluctuation amplitudes.

In summary, we have developed a confocal reflectance interferometric microscope that enables quantifying nanometer scale nucleic envelope and plasma membrane fluctuations in live cells with millisecond time resolution in non-contact and label-free fashion. The measured membrane fluctuations are directly related to cell's mechanical properties, which can be extracted by utilizing the measured temporal autocorrelation with an appropriate mathematical model. Label-free and non-contact quantification of nucleic mechanical properties will help understand key biological questions, such as the role of nuclear stiffness in cancer metastasis, especially during the extravasation process[18,55]. Finally, we believe that the proposed system, featuring fine depth selectivity and superior measurement sensitivity, will enable future studies of single cell biomechanics within tissues and animals in vivo.

## Methods

**Instrumentation**. The schematic of the confocal reflectance interferometric microscope is shown in Fig. 1. A frequency-doubled Nd-YAG laser operating at 532 nm (Verdi6 from Spectra Physics) is used as the light source. The pinhole array is created using a DMD-1, which allows simultaneous confocal imaging at multiple spatial locations. Diffraction-limited size pinholes are generated by turning 'ON' a set of DMD-1 micro-mirrors arranged on a grid. This pinhole array is relayed, using a 4-f system, to project the excitation field upon the specimen plane. The back-scattered optical field from the specimen is projected back to DMD-1, where only in-focus signals are passed through the ON-state micro-mirrors that also

function as an array of confocal pinholes to reject out-of-focus background. We note that DMD-1 can be reconfigured at a rate of over 20 kHz, allowing high-speed raster scanning of pinhole array for fast full-field imaging. The imaging speed is also inversely proportional to the total number of scanning patterns. Thus, for a given scanning speed of DMD, there is a trade-off between imaging speed and axial resolution as discussed in Supplementary Note 5.

A Zeiss ×40 objective with a numerical aperture (N.A.) of 1.2 (Zeiss C-Apochromat ×40/1.2 W Corr M27) is used for imaging. The physical size of each micro-mirror is 13.7 μm. We use 3 × 3 micro-mirrors to constitute a single confocal pinhole with an equivalent size of 1.02 μm at the specimen plane. A DLP Discovery 4100 kit (from Texas Instruments), which provides a binary pattern display rate up to 22 kHz using the 0.7XGA Chipset[41], is used to implement the scanning pinhole array. Wide field-of-view imaging is achieved by raster scanning the confocal pinhole pattern keeping two micro-mirror overlap. We used a total of 310 scanning patterns, which corresponds to wide-field imaging at ~70 Hz.

**Interferometric detection**. For interferometric imaging, the DMD-1 image plane is further relayed to a second imaging plane followed by interferometric detection. More specifically, a near common-path interferometer is implemented that shares the design with previously published diffraction phase microscopy[42]. To implement common-path interferometry, a grating is placed at the second image plane, which diffracts the signal beam into two orders (0 and +1) essentially generating two copies of the object beam. First lens (L4) of the second relay generates a frequency spectrum of the confocal-detected multi-aperture object field at the Fourier plane. A second DMD-2, DLP® lightcrafter™ evaluation module, is placed at this Fourier plane. Note that the light engine of the DLP lightcrafter module was removed and the chip with the micro-mirror array was placed at the Fourier plane. One half of DMD-2, corresponding to the 0th order, is programmed to act like a mirror by opening (or turning ON) all the micro-mirrors corresponding to that region. By allowing the entire spectrum corresponding to the 0th-order beam to pass, we generate the image of the object at the camera plane. To generate the reference wave for near common-path interferometry, a spatial frequency filter (COMB pattern) is generated on the remaining half of DMD-2. This COMB pattern is essentially the Fourier transform of the pinhole array used at the DMD-1 plane. This pinhole array filters out the high-frequency components of each COMB element of the first-order beam, effectively providing a reference wave with average phase of the object field. The object and reference beams interfere to form a complex interference pattern, which is recorded by a CMOS sensor (Hamamatsu ORCA 2.0). The theoretical framework describing the object and reference fields, and reconstruction of accurate quantitative phase corresponding to the specimen, is provided in detail in Supplementary Note 1.

**System calibration**. To demonstrate the depth resolution of the system, we scan a reflective surface along the axial direction. Briefly, interferograms of water–glass interface were recorded for different axial positions. The full width at half maximum (FWHM) of axial point spread function was determined as 1.5 μm, which defines the depth resolution of our system. Furthermore, a series of interferograms were also recorded to quantify the system's phase stability. The maximum phase noise observed from the system is around 5 milliradians that corresponds to about 160 pm. Finally, the accuracy of quantitative phase was measured by quantifying reflective phase specimen with 100 nm step height. See Supplementary Note 2 for details.

**Sample preparation**. All cell culture reagents were supplied from Invitrogen unless otherwise specified. Mouse E14 (ATCC, CRL-1821) ES cells were cultured in 5% $CO_2$/95% $O_2$ in DMEM supplemented with 10% ES cell qualified FBS, 2 mM L-glutamax, 1 mM sodium pyruvate, 100 μM non-essential amino acid, 0.1 mM β-mercaptoethanol, 1000 U mL$^{-1}$ mouse leukemia inhibitory factor (LIF, Millipore, ESG1106), and 100 U mL$^{-1}$ penicillin/100 μg mL$^{-1}$ streptomycin. For differentiation, 35 mm Petri dishes (Nunclon) were coated with fibronectin solution, where undifferentiated mouse ES cells were seeded at a moderate density (10$^3$ cells cm$^{-2}$) in culture medium without LIF.

**Reporting summary**. Further information on research design is available in the Nature Research Reporting Summary linked to this article.

## Data availability
The source data for all the figures are provided with the paper.

## Code availability
The codes used to analyze the source data are available on reasonable request from the corresponding author.

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

## Acknowledgements

The authors gratefully acknowledge the support by National Research Foundation, Prime Minister's Office, Singapore through the Singapore–MIT Alliance for Research and Technology's BioSystems and Micromechanics Inter-Disciplinary Research program (015824-039), and through Mechanobiology Institute (R-714-006-008-271); NMRC (R-185-000-294-511); NUHS Innovation Seed Grant (R-185-000-343-733); MOE ARC (R-185-000-342-112); IBN, IFCS & EMULSION, BMRC, A*STAR. National Institute of Health (NIH) (9P41EB015871-26A1, 5U01CA202177-02); and Hamamatsu Corporation.

## Author contributions

P.T.C.S., R.D.K., and H.Y. conceived the project. P.T.C.S., Z.Y., and V.R.S. proposed the idea and design of the system. V.R.S. built the set-up, performed experiments, analyzed the data, and prepared the figures. Y.A.Y. prepared the cells' samples and assisted during the experiments. V.R.S., P.T.C.S., and Z.Y. prepared the manuscript. All authors reviewed and edited the manuscript.

## Additional information

**Competing interests:** The authors declare no competing interests.

