## [Peer Review File · Nature Communications]

Reviewers' comments:

Reviewer #1 (Remarks to the Author):

The paper proposes a novel optical system for measuring fluctuations of inner organelles of biological cells. The system is based on rapid, full-field coherent illumination of the sample and parallel diffraction phase microscopy (DPM)-like system with dynamic pinhole array for obtaining sectioning in reflection mode. The optical system and the cell measurements are impressive. However, to be qualified for Nature Communications, you should be able to convince the readers that the measurements you provide are reliable. This can be done by either comparing to a gold standard method for mechanical measurements, or by taking three membranes with known mechanical properties, and show that the system can measure them together. Another verification, which is missing, is the complete sectioning capabilities of the proposed system. For this, you can use a transparent test target with several heights (like the USAF target you used but with more layers) and then show that you can completely reject out of focus layers with your method while profiling the different layers. Of course, these two verification experiments (mechanical properties and height profiling with sectioning) can be combined into a single experiment if you find the right phantom.

Reviewer #2 (Remarks to the Author):

This manuscript presents a new QPI approach, capable of extracting the nanoscale fluctuations of nuclear membranes, which can be converted into stiffness information. I find the development of this new confocal-phase imaging systems very powerful, with broad potential applications in biology. I have a few comments meant to clarify and hopefully improve the read, as follows.

1. There is some discussion about converting the fluctuations into mechanical information, but it turns out, no such analysis has been attempted. Since the word “mechanics” is announced in the title, I believe this would be important to try. I wonder if the model developed by Levine and used in Park et al. on red blood cell membranes may apply (1). This model takes as input the spatially-resolved correlation function (which the authors plot for one point in space) and outputs shear and bending moduli of the mebrane. At the minimum, the authors should discuss whether this description fits the experimental problem here.
2. The imaging system is new and exciting. However, it does share the common-path principle with diffraction phase microscopy (diffraction grating and pinholes), which went unacknowledged, it seems.
3. The confocal geometry is very useful for achieving good sectioning, but it also comes with a less obvious conversion of phase to thickness. The meaning of phase in backscattering measurements has been studies recently (2). Maybe the authors can comment on the underlying assumptions for using the simple formula

$$h(x, y; t) = \frac{\phi(x, y; t) \times \lambda}{4\pi \times n}$$

It might be that, because the confocal geometry porvides strong sectioning, the phase is the result of integration only over thin layers, in which case the formula applies.

4. In the abstract it is mentioned that QPM is powerful but “its lack of dept discrimination” prevented it from measuring cell mechanics beyond the red blood cell. It is true that QPM has not been used much for measuring mechanics of eukaryotic cells, but the statement that it lacks depth resolution is not universally accurate. As the authors no doubt know, there is a number of publications on QPI tomography (see, e.g., (3-5)).

5. It is interesting that the authors mention the disorder strength in the conetxt of tissues, but they missed the opportunity to discuss recent efforts in rertieving the disorder strenth thourgh QPI measurements (6, 7).

I believe that with these revisions, the manuscript will appeal to the broad readership of NComm.

References

1. Y. K. Park *et al.*, Measurement of red blood cell mechanics during morphological changes. *Proc. Nat. Acad. Sci.* **107**, 6731 (2010).
2. C. Hu, G. Popescu, Physical significance of backscattering phase measurements. *Optics Letters* **42**, 4643-4646 (2017).
3. Y. Cotte *et al.*, Marker-free phase nanoscopy. *Nat Photonics* **7**, 113-117 (2013).
4. W. Choi *et al.*, Tomographic phase microscopy. *Nature methods* **4**, 717 (2007).
5. T. Kim *et al.*, White-light diffraction tomography of unlabeled live cells. *Nat Photonics* **8**, 256-263 (2014).
6. W. J. Eldridge, Z. A. Steelman, B. Loomis, A. Wax, Optical Phase Measurements of Disorder Strength Link Microstructure to Cell Stiffness. *Biophysical Journal* **112**, 692-702 (2017).
7. M. Takabayashi, H. Majeed, A. Kajdacsy-Balla, G. Popescu, Disorder strength measured by quantitative phase imaging as intrinsic cancer marker in fixed tissue biopsies. *PLOS ONE* **13**, e0194320 (2018).
- 4.

Reviewer-1

The paper proposes a novel optical system for measuring fluctuations of inner organelles of biological cells. The system is based on rapid, full-field coherent illumination of the sample and parallel diffraction phase microscopy (DPM)-like system with dynamic pinhole array for obtaining sectioning in reflection mode. The optical system and the cell measurements are impressive. However, to be qualified for Nature Communications, you should be able to convince the readers that the measurements you provide are reliable. This can be done by either comparing to a gold standard method for mechanical measurements, or by taking three membranes with known mechanical properties, and show that the system can measure them together. Another verification, which is missing, is the complete sectioning capabilities of the proposed system. For this, you can use a transparent test target with several heights (like the USAF target you used but with more layers) and then show that you can completely reject out of focus layers with your method while profiling the different layers. Of course, these two verification experiments (mechanical properties and height profiling with sectioning) can be combined into a single experiment if you find the right phantom.

Responses:

We agree with the reviewer that the manuscript should provide information about the reliability of our measurements. In the proposed multi-point scanning confocal phase microscope, the reference field is generated by the specular reflection from a highly reflective coverglass interface, located at an off-focal plane, underneath the cell. This approach provides a highly phase-stabilized reference, which is critical for desired phase measurement stability. However, the presence of strong signal from the glass interface also suppresses the measured phase associated with the fluctuating membrane. Previous literature suggests that the presence of strong reflector, in close proximity to the interface of interest, leads to suppressed phase measurement in low-coherence depth-resolved optical imaging systems [1]. We note that the approach presented in Ref [1] is based on point-illumination, where several measurements are needed to account for phase leakage. We thank the reviewer for raising this important point, and we have developed and added a systematic approach to obtain accurate phase measurements by acquiring a single wide-field interferogram. In the following, we briefly describe the theoretical model, simulations, and experimental results presenting membrane fluctuations in healthy red blood cells (RBCs). For validation, we also measured RBC membrane fluctuations using two additional established methods as “gold standards”, namely, the diffraction phase microscopy (DPM) [2] and dynamic speckle illumination phase microscopy [3]. This work is reported as an additional section, S3, in the supplementary material.

Modeling the suppression of membrane phase fluctuations due to the presence of reflection signal from glass interface: Let E_c and E_0 are the amplitudes of signals from in-focus cellular membrane and off-focus glass interface respectively. ϕ_0 and ϕ_c^0 represent the mean phase

associated with light arriving from glass interface and the cell membrane. Furthermore, $\phi(t)$ is the temporal phase fluctuations of the membrane under observation; glass interface is assumed to be static. The total back scattered field, E_T , can be written as,

$$E_T = E_0 e^{i\phi_0} + E_c e^{i\{\phi_c^0 + \phi(t)\}} \quad (1)$$

Figure 1: Suppression in measured phase associated with total field as a function of optical signal from the fluctuating membrane and contribution from the glass interface.

Figure 1 shows the plot of rms phase of the total backscattered field (red trace) as a function of contribution from glass interface. Clearly, the phase is accurate when there is no contribution from glass interface. On the other hand, the phase fluctuation is suppressed when signal from glass contributes to the total field; however, the amount of suppression is deterministic. A common-path interferometer is placed in the detection arm of the system to evaluate the phase of the total field. The interference of object and reference fields is recorded at the camera plane, where the object field is the same as the total field as defined in Eq. 1 and the reference field is generated from the total field using a customized filter (see supplementary section S1). For our near-common path system, the generation of reference field at any location on the camera is the spatial average of the total field detected from all the foci in the field-of-view (see Eq. 3 in the supplementary material, S1). Thus, for the total object field defined in Eq. 1, the reference field generated at the camera in our system can be written as $E_R = \overline{E}_0 e^{i\overline{\phi}_0} + \overline{E}_c e^{i\overline{\phi}_c}$, where \overline{E}_0 and \overline{E}_c are the spatially averaged amplitudes of optical signal contributions from the glass and the membrane interfaces, respectively. Further, $\overline{\phi}_0$ and $\overline{\phi}_c$ are their corresponding spatially averaged phase shifts. Since \overline{E}_c depends on the fractional area covered by the cells in the field-of-view (FOV), the generation of reference field is completely governed by the cell membrane contributions in absence of signal from glass interface. In such a scenario,

the variation in cell density within the FOV will lead to a non-stable reference field. For instance, rms phase of fluctuating membrane is not recovered at all for 10% or less cell density for in the absence glass contribution (see the first data point of yellow trace in Fig. 1). However, as the glass contribution increases, rms phase of the total field starts to recover. For larger than 10% cell densities, rms phase can be measured without any glass contribution, but not without significant error. Thus, a stable reference field from the glass interface, with similar or larger amplitude than that of the fluctuating membrane, is advantageous for the recovery of accurate phase fluctuation information. In our experiments, the contribution of glass interface (normalized by the membrane signal) is above 2, which is essential for stable phase recovery.

As discuss earlier, while the glass interface signal ensures a stable reference field, its static nature also results in “phase suppression.” In the following, we describe a model for correcting the suppressed phase. For off-axis interferometry, the reference field can be defined as

$$E_R = E_r e^{i\phi_r} e^{ik_x}, \quad (2)$$

where e^{ik_x} is the off-axis phase tilt. The detected interferogram can be written as

$$\begin{aligned} I &= (E_T + E_R) \times (E_T + E_R)^* \\ &= \\ E_0^2 + E_r^2 + E_c^2 + 2E_0E_c \cos(\phi_0 - \phi_c^0 - \phi(t)) &+ 2E_rE_0 \cos(\phi_r + k_x - \phi_0) + 2E_rE_c \cos(k_x + \\ \phi_r - \phi_c^0 - \phi(t)) & \end{aligned} \quad (3)$$

Next, the interferogram is processed after taking its Fourier transform:

$$\begin{aligned} \tilde{I} &= [E_0^2 + E_r^2 + E_c^2 + 2E_0E_c \cos(\phi_0 - \phi_c^0 - \phi(t))] \delta(k_x) + E_rE_0 (\delta(k+k_x) + \delta(k-k_x)) e^{i(\phi_0 - \phi_r)} \\ &+ E_rE_c (\delta(k+k_x) + \delta(k-k_x)) e^{i(\phi_c^0 + \phi(t) - \phi_r)} \end{aligned}$$

The phase directly reconstructed from off axis interferogram, using Hilbert transform, can be written as,

$$\varphi = \tan^{-1} \left(\frac{E_rE_0 \sin(\phi_0 - \phi_r) + E_rE_c \sin(\phi_c^0 + \phi(t) - \phi_r)}{E_rE_0 \cos(\phi_0 - \phi_r) + E_rE_c \cos(\phi_c^0 + \phi(t) - \phi_r)} \right) \quad (4)$$

As discussed earlier, the direct phase recovered from interferogram is suppressed due to presence of signal from glass interface, i.e., E_0 . To recover the corrected phase, the DC and AC component of the interferogram can be written as:

$$I_{DC} = E_0^2 + E_r^2 + E_c^2 + 2E_0E_c \cos(\phi_0 - \phi_c^0 - \phi(t)) \quad (5)$$

$$I_{AC} = E_rE_0 e^{i(\phi_0 - \phi_r)} + E_rE_c e^{i(\phi_c^0 + \phi(t) - \phi_r)} \quad (6)$$

For outside the cell region, magnitude of the DC is $E_0^2 + E_r^2$ and magnitude of the AC is $E_r E_0$. By using the measured DC and AC values and solving the algebraic equations, one can evaluate the values of E_r and E_c .

Finally, by rearranging the AC term, the recovered phase can be calculated as

$$\phi_r - \phi_c^0 - \phi(t) = \tan^{-1} \left(\frac{Im(I_{AC}) - E_r E_0 \sin(\phi_r - \phi_0)}{Re(I_{AC}) - E_r E_0 \cos(\phi_r - \phi_0)} \right), \quad (7)$$

where $Re(I_{AC})$ and $Im(I_{AC})$ are the real and imaginary part of AC, respectively, inside the cell region.

Experimental validation: Membrane fluctuations of healthy red blood cell (RBC) are measured for experimental validation. The phase of the membrane is converted to the height followed by the rms fluctuation measurements, which are further compared with the one measured using the diffraction phase microscopy (DPM) [2] and dynamic speckle phase microscopy [3] systems for the same RBC population. DPM system is a well-known transmission-type near-common path wide-field interferometric tool for measuring nanometer scale thermally driven RBCs membrane fluctuations [4, 5]. On the other hand, dynamic speckle phase microscope is a recently developed wide-field reflection-type interferometric tool that provides depth-resolved quantitative phase maps of biological samples similar to the current manuscript with the exception that it has a non- common path design. It has been shown to selectively measure top membrane fluctuations in RBCs [3]. We believe that comparing the RBC membrane fluctuations, measured using the proposed system, with the above-mentioned two modalities should provide a fair assessment of accuracy of the system.

Figure 2: Recovery of suppressed phase for red blood cell membrane fluctuations. (a)-(c) show the phase map of RBC, its rms phase fluctuation map, and instantaneous height map (the scale bar is in nm) using Hilbert transform. (d)-(f) represent the corresponding corrected phase image, rms phase fluctuations and instantaneous height map, respectively, using the proposed model [see Eq. (7)]. (g) Comparison of original and corrected temporal phase fluctuations of the cell membrane. (h) Comparison of membrane rms height fluctuations measured using the presented confocal interferometric microscopy, dynamic speckle based phase, and DPM systems, respectively.

First, we record a series of interferograms using the proposed system by keeping the focal plane aligned with the top membrane of RBC. Figure 2(a) shows the phase image of a RBC directly reconstructed from the measured interferogram using Hilbert transform. Figures 2(b) and 2(c) illustrate the corresponding rms fluctuation and instantaneous displacement maps, respectively. We observe a typical value of ~ 0.15 radian rms phase, which corresponds to ~ 5 nm membrane fluctuation amplitude. This value is significantly lower than that measured with DPM (~ 35 nm) or dynamic speckle phase microscope (~ 21 nm). We note that higher rms fluctuation amplitude is observed with DPM (than dynamic speckle) due to its transmission geometry, which records fluctuations of both top and bottom membranes of the RBC. Next, we use the proposed reconstruction model [Eq. (7)] to compute the phase maps as well as the instantaneous displacement and rms fluctuation maps, as shown in Figs. 2(d-f). As expected, the recovered rms phase fluctuations was higher and measured to be ~ 0.6 radians. An increase of ~ 4 times in rms phase is also expected since the amplitude of signal from glass interface was about 1 to 2 times higher than the amplitude of signal from the cell membrane, which points to a phase suppression of ~ 4 times (see Fig. 1). A comparison of original and corrected temporal phase fluctuations measured at a single spatial location is also shown in Fig 2(g). Further, the measurements were extended to a number of RBCs ($n = 7$) using the proposed confocal reflectance interferometric system. For comparison, the measurements of membrane fluctuations of the same RBC population were also performed using the dynamic speckle phase microscopy and the DPM systems. More precisely, the rms fluctuation amplitude was measured at 4 different spatial locations on each RBC. As shown in Fig. 2(h), the rms fluctuations measured using the proposed system are comparable with that measured using dynamic speckle phase system. This is due to the fact that both the systems essentially provide depth-resolved phase measurement in back scattered mode. We note that the rms fluctuation measured using dynamic speckle based system is slightly higher, which may be attributed to higher phase noise because of its non-common path geometry. Further, approximately $\sqrt{2}$ higher rms fluctuation amplitude is expected for DPM due to its transmission geometry, which measures fluctuation contributions from both top and bottom membranes.

Validation of optical sectioning capability: Although Fig. S2(a) in supplementary material presents the experimentally measured axial point spread function, representing the sectioning capability of the system, we further demonstrate this capability using an additional experiment as follows. A sample is prepared by using two glass coverslips attached to each other using a double sided tape (~ 30 microns thick) used as a spacer, as shown in Fig. 3(a). Imaging of the top and bottom interfaces was performed using the proposed system. Figures 3(b) and 3(c) show the intensity images corresponding to the bottom and top interfaces, respectively. Clearly, axial

sectioning capability of the system is observed, when no contribution from the bottom interface is observed while imaging the top interface. It is further clearly shown by quantitatively comparing the intensities of different region in Fig. 3(d).

Figure 3: (a) Schematic of the two cover slides used for imaging, red region showing the imaging field of view. (b) Image recorded at the bottom interface, (c) interferogram recorded at the top interface, and (d) intensity comparison of different regions

Reviewer-2

This manuscript presents a new QPI approach, capable of extracting the nanoscale fluctuations of nuclear membranes, which can be converted into stiffness information. I find the development of this new confocal-phase imaging systems very powerful, with broad potential applications in biology. I have a few comments meant to clarify and hopefully improve the read, as follows.

1. There is some discussion about converting the fluctuations into mechanical information, but it turns out, no such analysis has been attempted. Since the word “mechanics” is announced in the title, I believe this would be important to try. I wonder if the model developed by Levine and used in Park et al. on red blood cell membranes may apply (1). This model takes as input the spatially-resolved correlation function (which the authors plot for one point in space) and outputs shear and bending moduli of the membrane. At the minimum, the authors should discuss whether this description fits the experimental problem here.

Response: The reviewer is correct that we have not explicitly measured the rheological moduli of the interfaces although our past experience has shown that interface fluctuation amplitude is a very useful surrogate for many biomedical applications [6]. While the path towards recovery of rheological parameters is relatively straight forward, we believe that this work is beyond the scope of the current paper because it involves many steps beyond developing novel optical instrumentation that is our focus. Specifically, the model developed by Levine and used by Park

et al. interprets the measured membrane fluctuations using a viscoelastic continuum model of the composite spectrin-network/lipid membrane bounded by bulk viscous fluids on both sides. While this is a good model for red blood cells, it is clearly insufficient for eukaryotic cells. For eukaryotic cells, the interior of the cell is not homogenous as RBCs but has at least two major compartments, namely, the cytosol and the nucleus, with different viscoelastic properties. Therefore, the simplest model must consist of three compartments: medium (pure viscous fluid), cytosol (a viscoelastic medium), nucleus (a different viscoelastic medium) bounded by two interfaces (plasma membrane and nucleus membrane) with different bending, κ , and extension, σ , moduli. Granek [7] has reported a relevant model that can recover bending modulus of a membrane driven by thermal forces separating two viscoelastic media with different frequency-dependent complex moduli using spatial and temporal autocorrelation functions of the interface fluctuations that can be quantified using confocal phase microscopy presented here. Since we expect that extension moduli to dominate in the spatial/temporal scales of eukaryote cells measured in millisecond to second time scale, the Granek model will need to be extended theoretically to account for both the bending and stretch moduli; this extension is fairly simple given these two moduli have very different spatial and temporal power dependence [7]. Finally, the Granek model can then be extended for the three-compartment case to determine plasma and nucleic membrane moduli assuming the viscoelastic properties of the cytosol and the nucleus can be independently measured. Recent development of force spectrum microscopy [8], an advanced form of laser trap activated particle rheology, offers an attractive approach to accurately determine the unknown frequency dependent viscoelastic properties of these two compartments. Future experiments are planned to incorporate force spectrum microscopic measurement in model systems to validate the recovery of plasma and nucleic membrane bending and extension moduli based on confocal phase microscope measurements of their fluctuation amplitudes. This explanation has been added to the discussion section of the paper.

2. *The imaging system is new and exciting. However, it does share the common-path principle with diffraction phase microscopy (diffraction grating and pinholes), which went unacknowledged, it seems.*

Response: We agree that the presented interferometer design is similar to diffraction phase microscopy; however, the reference path is quite different and it is explained in the supplementary section S1. We have included the following addition in revised manuscript (page 6):

“...a highly stable near common-path interferometer, which shares the design with previously published diffraction phase microscopy [43], is implemented in the detection arm to quantify...”.

2. *The confocal geometry is very useful for achieving good sectioning, but it also comes with a less obvious conversion of phase to thickness. The meaning of phase in backscattering measurements has been studied [studied] recently (2). Maybe the authors can comment on the underlying assumptions for using the simple formula $h(x, y; t) = \frac{\phi(x, y; t) \times \lambda}{4\pi n}$, It might be*

that, because the confocal geometry provides strong sectioning, the phase is the result of integration only over thin layers, in which case the formula applies.

Response: We agree that the given formula is primarily valid for converting phase to thickness from back scattered signal for plane wave illumination. In the proposed confocal geometry, DMD-1 plane (flat surface) provides a flat phase for the excitation field at the focal region within the specimen. The phase of backscattered field should be measured against this reference (as shown in the simulation results in the supplementary material, S1). As pointed out by the reviewer, our system also has strong optical sectioning (1.5 microns), which warrants the use of above expression to convert phase into axial displacements. We also thank the reviewer to point out a relevant reference. We have now updated our references by including the article in supplementary material.

4. In the abstract it is mentioned that QPM is powerful but “its lack of dept[h] discrimination” prevented it from measuring cell mechanics beyond the red blood cell. It is true that QPM has not been used much for measuring mechanics of eukaryotic cells, but the statement that it lacks depth resolution is not universally accurate. As the authors no doubt know, there is a number of publications on QPI tomography (see, e.g., (3-5)).

Response: We agree with the reviewer’s comment and have revised the abstract. Specifically, we have added the following text:

“... and has been successfully used to study red blood cell rheology. It, however, has not been utilized to study biomechanics of complex eukaryotic cells either due to lack of depth sectioning, limited phase measurement sensitivity, or both.”

5. It is interesting that the authors mention the disorder strength in the conetxt [context] of tissues, but they missed the opportunity to discuss recent efforts in rertieving [retrieving] the disorder strenth thourgh [strength through] QPI measurements (6, 7).

Response: We thank the reviewer to point out the recent work on retrieving the disorder strength parameter through quantitative phase measurements. We have added the following information in the revised manuscript:

“Recently, quantitative phase microscopy (QPM) based methodologies have also been presented to estimate the disorder strength, which can be related to the refractive index variance in biological cells and tissue samples. More precisely, the measured disorder strength parameter has been linked with the cell stiffness [23] and used to compare benign versus malignant breast tissue biopsies [24].”

References:

1. A.K. Ellerbee et. al., “Phase retrieval in low-coherent interferometric microscopy”, *Opt. Lett.*, Vol. 32, 388-390, 2007.
2. G. Popescu et al., “Diffraction phase microscopy for quantifying cell structure and dynamics”. *Opt. Lett.*, Vol. 31: p. 775-777, 2006.
3. Y. Choi et. al., “Dynamic speckle illumination wide-field reflection phase microscopy”, *Opt. Lett.*, Vol. 39, 6062-6065, 2014.
4. YK Park et. al., “Quantitative phase imaging in biomedicine”, *Nature Photonics*, Vol. 12, 578-589, 2018.

5. T. Ling et. al., "Full-field interferometric imaging of propagating action potentials", *Light: Science & Applications*, 7: 107, 2018.
6. P. Hosseini, "Cellular normoxic biophysical markers of hydroxyurea treatment in sickle cell disease", *PNAS*, Vol. 113, 9527-9532, 2006.
7. Granek, R., "Membrane surrounded by viscoelastic continuous media: anomalous diffusion and linear response to force", *Soft Mat.*, Vol 7, 5281- 5289, 2011.
8. M. Guo et.al., "Probing the stochastic, motor-driven properties of the cytoplasm using force spectrum microscopy", *Cell*, Vol. 158, 822-832, 2014.

REVIEWERS' COMMENTS:

Reviewer #1 (Remarks to the Author):

The authors performed my previous round remarks. I support publication.

NTS

Reviewer #2 (Remarks to the Author):

The authors addressed adequately all my concerns.

Reviewer-1

The authors performed my previous round remarks. I support publication.

Responses:

We thank the reviewer to perform review of our revised manuscript and happy to hear that our revision addressed the remarks of the reviewer.

Reviewer-2

The authors addressed adequately all my concerns [concerns].

Responses:

We thank the reviewer to perform review of our revised manuscript and happy to hear that our revision addressed the concerns of the reviewer.